# Long-Lived States Involving a Manifold of Fluorine-19 Spins in Fluorinated Aliphatic Chains

Coline Wiame, Sebastiaan Van Dyck, Kirill Sheberstov, Aiky Razanahoera and Geoffrey Bodenhausen Chimie Physique et Chimie du Vivant (CPCV, UMR 8228), Department of Chemistry, Ecole Normale Supérieure, PSL University, Sorbonne University, CNRS, Paris, 75005, France

Correspondence to: Kirill Sheberstov (kirill.sheberstov@ens.psl.eu)

**Abstract.** Long-lived states (LLS) have lifetimes  $T_{\rm LLS}$  that exceed longitudinal spin-lattice relaxation times  $T_1$ . In this study, lifetimes  $T_{\rm LLS}(^{19}{\rm F})$  have been measured in three different achiral per- and polyfluoroalkyl substances (PFAS) containing 2 or 3 consecutive CF<sub>2</sub> groups. In a static magnetic field  $B_0 = 11.7$  T, the lifetimes  $T_{\rm LLS}(^{19}{\rm F})$  exceed the longitudinal relaxation times  $T_1(^{19}{\rm F})$  by about a factor 2. The lifetimes  $T_{\rm LLS}(^{19}{\rm F})$  can be strongly affected by binding to macromolecules, a feature that can be exploited for screening of fluorinated drugs. Both  $T_{LLS}(^{19}{\rm F})$  and  $T_1(^{19}{\rm F})$  should be longer at lower fields where relaxation due to the chemical shift anisotropy (CSA) of  $^{19}{\rm F}$  is less effective, which is demonstrated here by running experiments at both fields of 11.7 T and 7 T.

## 1 Introduction

The discovery of long-lived states (LLS) was inspired by spin isomers of dihydrogen (H<sub>2</sub>), in which a nonequilibrium ratio of para- vs ortho-dihydrogen can persist for many days before relaxing. Many applications of LLS have been illustrated for pairs of <sup>13</sup>C or <sup>15</sup>N nuclei (Pileio et al., 2012a; Feng et al., 2013; Stevanato et al., 2015; Elliott et al., 2019; Sheberstov et al., 2019b), for pairs of <sup>1</sup>H nuclei (Franzoni et al., 2012; Kiryutin et al., 2019a), and less frequently, for pairs of <sup>19</sup>F nuclei (Buratto et al., 2016), for <sup>31</sup>P (Korenchan et al., 2021), and for <sup>103</sup>Rh spins (Harbor-Collins et al., 2024). The slow decays of LLS have been exploited for the determination of small diffusion coefficients (Cavadini et al., 2005), for measurements of slow chemical exchange (Sarkar et al., 2007), for the storage of hyperpolarization (Vasos et al., 2009; Pileio et al., 2012b; Kiryutin et al., 2019b), for investigations of weak ligand-protein binding (Salvi et al., 2012), and for studies of tortuosity in porous media (Dumez et al., 2014; Pileio et al., 2015; Pileio and Ostrowska, 2017; Tourell et al., 2018; Melchiorre et al., 2023). Another application of LLS is spectral editing, in particular for selective filtering of selected signals in proteins (Mamone et al., 2020; Sicoli et al., 2024).

#### 1.1 Fluorinated Drugs

Many drugs contain one or more fluorine atoms since pharmacokinetic studies have shown that their lifetimes *in vivo* are favored because C-F bonds are harder to break down by enzymes than C-H bonds (Shah and Westwell, 2007). Positron

emission tomography (PET) with <sup>18</sup>F-labeled radioligands is widely applied in diagnosis and in drug development (Nerella et al., 2022). Fluorinated drugs also have the advantage of being easy to track by NMR and MRI, since the sensitivity of <sup>19</sup>F NMR is comparable to that of <sup>1</sup>H NMR, while signal overlap is significantly reduced due to the greater chemical shift dispersion, and spectral quality is improved because of the absence of background signals, e.g., water signal (Buchholz and Pomerantz, 2021). For drug screening, it is important to achieve the best possible contrast between the response of a ligand L that binds to a target protein P and a molecule that fails to do so. A LLS involving two or more <sup>19</sup>F spins in a ligand can lead to a remarkable contrast upon binding to a protein target (Buratto et al., 2016). For ligands that bind to a target protein, contrast in <sup>19</sup>F NMR arises from the combined effects of binding on the chemical shifts, on the reduction of symmetry when achiral ligands bind to chiral targets, and on the correlation times of rotational diffusion. The latter effect leads to line-broadening by homogeneous transverse *T*<sub>2</sub> relaxation. In addition, chemical exchange in the intermediate regime can also contribute to line-broadening (Buchholz and Pomerantz, 2021). The latter two effects can be distinguished by comparing Carr-Purcell-Meiboom-Gill (CPMG) echo trains with high repetition rates and slow CPMG echo trains using so-called "perfect echoes". The first type of experiment will suppress echo modulations due to homonuclear scalar couplings "*J*(<sup>19</sup>F, <sup>19</sup>F) and inhibit echo decays due to intermediate exchange (Takegoshi et al., 1989; Aguilar et al., 2012), while the second type of experiment eliminates the effects of intermediate exchange (Lorz et al., 2025).

## 1.2 Long-Lived States

30

In a pair of two spins A and A', the difference between the population of the singlet state  $p(S_0^{AA'})$  and the mean population of the three triplet states  $\langle p(T^{AA'}) \rangle = \frac{1}{3} (p(T_{+1}^{AA'}) + p(T_0^{AA'}) + p(T_{-1}^{AA'}))$  is known as *triplet-singlet population imbalance*, which is immune against relaxation driven by intra-pair dipole-dipole couplings. One speaks of LLSs since these population imbalances decay with lifetimes  $T_{LLS}$  that can greatly exceed  $T_1$  and  $T_2$ . Such a state can be described by a scalar product  $\hat{I}^A \cdot \hat{I}^{A'}$ , where  $\hat{I}^P = (\hat{I}_x^P, \hat{I}_y^P, \hat{I}_z^P), p \in \{A, A'\}$ . In an achiral aliphatic chain with 6 spins, denoted to AA'MM'XX' in Pople's notation, one can excite not only two-spin order terms  $\hat{I}^A \cdot \hat{I}^{A'}$  and/or  $\hat{I}^M \cdot \hat{I}^{M'}$ , and/or  $\hat{I}^X \cdot \hat{I}^{X'}$  but also 4-spin order terms such as  $(\hat{I}^A \cdot \hat{I}^{A'}) \cdot (\hat{I}^M \cdot \hat{I}^{M'})$ , and/or  $(\hat{I}^A \cdot \hat{I}^{A'}) \cdot (\hat{I}^X \cdot \hat{I}^{X'})$ , although simulations for protonated aliphatic chains indicate that the yields of such a 6-spin term are low (Sonnefeld et al., 2022a). It is challenging to determine separately the coefficients if one excites admixtures of all seven possible products in a 6-spin system. Since experiments indicate that the decay constants are often very similar, admixtures of different products tend to relax with an effective mono-exponential decay. These remarks apply to molecules containing either three fluorinated or three protonated methylene groups, including those analyzed in this study and in our previous work on protons (Sonnefeld et al., 2022a).

In many molecules, there are two <sup>19</sup>F spins that have different chemical shifts, e.g., in many di-fluoro-substituted aromatic rings. In *chiral* molecules containing CF<sub>2</sub> groups, the diastereotopic <sup>19</sup>F nuclei have different chemical shifts, if they are not

too far from a stereogenic centre. In such cases, it is straightforward to excite and observe an LLS involving the two <sup>19</sup>F spins of a CF<sub>2</sub> group. In our laboratory, we often use a sequence of hard pulses developed for this purpose (Sarkar et al., 2007). LLSs have thus been observed in diastereotopic CF<sub>2</sub> groups where their lifetimes  $T_{LLS}$  exceed  $T_I$  significantly (Buratto et al., 2016). In *achiral* molecules on the other hand, pairs of <sup>19</sup>F atoms attached to the same carbon are chemically equivalent, i.e., have degenerate chemical shifts, so that the geminal  $^2J$  ( $^{19}F$ ,  $^{19}F$ ) couplings does not affect the spectra to first order. Yet LLS can be excited in such systems provided that the pairs of <sup>19</sup>F atoms are *magnetically inequivalent*. A pair of <sup>19</sup>F atoms attached to the same carbon atom are magnetically inequivalent if and only if vicinal couplings such as  $^3J$ ( $^{19}F$ ,  $^{19}F$ ), or  $^3J$ ( $^{19}F$ ,  $^{1}H$ ), to  $^{19}F$  or  $^{1}H$  nuclei in neighbouring CF<sub>2</sub> or CH<sub>2</sub> groups are *not* degenerate, i.e., provided differences such as  $\Delta J_{AM} = J_{AM} - J_{AM'} = J_{A'M'} - J_{A'M}$  and  $\Delta J_{MX} = J_{MX} - J_{MX'} = J_{M'X'} - J_{M'X}$  do not vanish. The degeneracy of vicinal scalar couplings is lifted when the rotamers produced by rotations about the C–C bonds are not equally populated. This occurs if the potential wells corresponding to the different rotamers have unequal energy.

This work extends the excitation of LLS by Spin-Lock Induced Crossing (SLIC) (DeVience et al., 2013) to a set of 4 or 6  $^{19}$ F spins in perfluorinated aliphatic chains of the type  $-(CF_2)_n$ . Specifically, we have looked at per- and polyfluoroalkyl (PFAS) molecules, which have been widely studied for their detrimental effects on the environment (Fenton et al., 2021). Studying these molecules and their ability to bind to protein targets could shed more light on their impact on living organisms.

Excitation with a SLIC pulse can also enhance the intensity of Outer Singlet–Triplet (OST) coherences (Sheberstov et al., 2019a), as discussed in more detail below. We have used mono- and poly-chromatic SLIC, involving the simultaneous application of 1, 2 or 3 radio-frequency (RF) fields to 1, 2 or 3 multiplets in a  $^{19}$ F spectrum (Fig. 1), in analogy to our work on  $^{1}$ H spins in aliphatic chains of the type –(CH<sub>2</sub>)<sub>n</sub> – (Sonnefeld et al., 2022a, b; Razanahoera et al., 2023; Sheberstov et al., 2024). The resulting long-lived states involve 2, 4 or, 6  $^{19}$ F spins. One must distinguish between two approaches: single- or double-quantum SLIC (SQ- or DQ-SLIC). For DQ-SLIC to be efficient, the RF amplitudes  $v_{SLIC}$  of SLIC pulses must be equal to the geminal coupling  $^{2}J$  ( $^{19}$ F,  $^{19}$ F) between two neighbouring  $^{19}$ F spins, or twice as large for SQ-SLIC.

Figure 1. (a) Experiment used to amplify the amplitudes of forbidden "Outer Singlet-Triplet Transitions" (OSTs) in conventional  $^{19}$ F spectra. The radio-frequency (RF) amplitude  $v_{SLIC}$  and duration  $\tau_{SLIC}$  can be optimized empirically to achieve the highest possible

OST signal amplitudes (Sheberstov et al., 2019a). (b) Pulse sequence used to study the excitation, relaxation, and reconversion of LLS of <sup>19</sup>F in fluorinated achiral aliphatic chains. The transverse magnetization is excited by a 'hard' non-selective (π/2)<sub>x</sub> pulse, followed by the application of one, two or three selective RF fields (polychromatic SLIC pulses) applied simultaneously at the resonance frequencies (chemical shifts) of one, two or three consecutive CF<sub>2</sub> groups to convert the magnetization into a superposition of various LLS. Two optima result from the level anti-crossings at the single-quantum condition (SQ LAC) or at the double-quantum condition (DQ LAC). The RF amplitudes must be equal to the geminal coupling for DQ LAC, i.e., v<sub>SLIC</sub><sup>DQ</sup> = <sup>2</sup>J (<sup>19</sup>F, <sup>19</sup>F), or twice as large for SQ LAC. The maximum efficiency is achieved either for a short pulse duration τ<sup>SQ</sup><sub>SLIC</sub> = 1/(|√2ΔJ|) for SQ LAC, or a longer duration τ<sup>DQ</sup><sub>SLIC</sub> = √2τ<sup>SQ</sup><sub>SLIC</sub> = 1/(|ΔJ|) for DQ LAC, where ΔJ = (ΔJ<sub>AM</sub> + ΔJ<sub>MX</sub>)/2 and ΔJ<sub>AM</sub> = J<sub>AM</sub> - J<sub>AM</sub> - J<sub>A'M</sub> - J<sub>A'M</sub> and ΔJ<sub>AX</sub> = J<sub>AX</sub> - J<sub>AX</sub> - J<sub>A'X</sub> . After a T<sub>00</sub> filter (Tayler, 2020), another set of mono- or poly-chromatic SLIC pulses allows one to reconvert LLS into observable magnetization.

A population imbalance between the triplet and singlet states can also be obtained at very low spin temperatures, as may occur in dynamic nuclear polarization (DNP) (Tayler et al., 2012; Bornet et al., 2014; Razanahoera et al., 2024). In systems with more than two spins, one can also excite long-lived imbalances between states that belong to different symmetries of the spin permutation group, e.g. an imbalance between populations associated with irreducible representations A and E in CD<sub>3</sub> groups (Kress et al., 2019).

#### 105 2 Results and Discussion

To extend the excitation of LLS from  $^{1}$ H to  $^{19}$ F, a challenge arises from the fact that  $J(^{19}\text{F}^{-19}\text{F})$  and  $J(^{1}\text{H}^{-1}\text{H})$  differ in the way their magnitude depend on the number of chemical bonds between atoms. Typically, geminal  $^{2}J(^{19}\text{F}^{-19}\text{F})$  couplings in CF<sub>2</sub> groups are on the order of 250 to 290 Hz (Krivdin, 2020), much larger than geminal  $^{2}J(^{1}\text{H}^{-1}\text{H})$  couplings in CH<sub>2</sub> groups which are about  $\sim$  -15 Hz. In  $^{1}$ H NMR, typical values of the difference between vicinal couplings lie in the range  $0 < \Delta J(^{1}\text{H}) < 7$  Hz, while for  $^{19}$ F NMR, these differences may typically lie in the range  $0 < \Delta J(^{19}\text{F}) < 40$  Hz.

We have explored three achiral fluorinated molecules shown in Fig. 2, selected because the chemical shifts between neighboring CF<sub>2</sub> groups exceed 1 kHz. This means that LLS excitation of a chosen CF<sub>2</sub> group is sufficiently selective, even with RF amplitudes on the order of 250 to 290 Hz for DQ LAC, or on the order of 500 to 580 Hz for SQ LAC respectively, so that one can neglect the effects of a SLIC pulse on neighboring CF<sub>2</sub> groups.

Figure 2. Three achiral molecules with fluorinated aliphatic chains studied in this work: perfluorobutanoic acid (PFBA), perfluorobutane sulfonic acid (PFBS), and perfluoropentanoic acid (PFPeA). All three molecules were dissolved in DMSO-d6 at 500 mM concentrations. The spectra were acquired at 298 K on a Bruker WB spectrometer at 11.7 T (470.46 MHz for <sup>19</sup>F, 500 MHz for <sup>1</sup>H) equipped with a NEO console.

## 2.1 Outer Singlet-Triplet Transitions

For the excitation of LLS in aliphatic chains containing  ${}^{1}H$  (Sonnefeld et al., 2022b), we have estimated the geminal  ${}^{2}J({}^{1}H-{}^{1}H)$  couplings and used spin simulation programs (Cheshkov et al., 2018) to optimize the radio-frequency field amplitude  $v_{SLIC}$  and the duration  $\tau_{SLIC}$ . For  ${}^{19}F$ , it is more difficult to estimate all relevant coupling constants. As a consequence of chemical

equivalence, the geminal couplings  ${}^2J$  ( ${}^{19}F$ ,  ${}^{19}F$ ) cannot be observed as a splitting, but manifest themselves through weakly allowed combination lines that appear on either side of the  ${}^{19}F$  multiplets with intensities that are typically  $10^4$  times weaker than those of the allowed transitions. The detection of the OST transitions can be improved (Sheberstov et al., 2019a) by irradiating one of the multiplets with a single monochromatic RF field with an amplitude  $v_{rf}$  in the vicinity of the optimum value  $v_{SLIC}$  and a duration near the ideal  $\tau_{SLIC}$  (Fig. 1a.) Thus, we were able to enhance the intensities of the OST transitions by up to two orders of magnitude. In this work, we have in this manner measured geminal couplings that fall in the range of  $280 

Figure 3. a) Conventional <sup>19</sup>F spectrum at 11.7 T (470.46 MHz for <sup>19</sup>F) of the multiplet of the low-frequency CF<sub>2</sub> group (peak 3 in Figure 2) of 6.9 M perfluorobutanoic acid (PFBA) in DMSO-d6. The weakly allowed combination lines, known as Outer Singlet-Triplet transitions (OSTs), are emphasized by rectangular frames, but cannot be reliably identified as such on the grounds of the top spectrum alone. b) The OSTs were enhanced by irradiation with an RF field with an amplitude  $v_{SLIC} = 574$  Hz, applied to the centre of the low-frequency CF<sub>2</sub> group in PFBA during  $\tau_{SLIC}$  50 ms, immediately followed by the observation of the <sup>19</sup>F free induction decay (i.e., without conversion into LLS). The frequency difference between the two framed transitions is 1148 Hz, which

corresponds to four times  ${}^2J({}^{19}F^{-19}F)$ . The top spectrum required 1024 scans, while the bottom spectrum was obtained with only 4 scans.

## 2.2 Lifetimes of Long-Lived States

The preliminary experiments shown in Fig. 3 allowed us to optimize conditions for SLIC. Using the pulse sequence of Fig. 1b, we observed the decay curves of Fig. 4.

Figure 4. Decays of LLS (actually admixtures of two-, four-, and minor amounts of six-spin order LLS terms) for three different fluorinated molecules at 11.7 T (470.46 MHz for <sup>19</sup>F). The decays were fitted with mono-exponential functions to determine the LLS lifetimes reported in Table 1. The CF<sub>2</sub> groups for which the decays are shown are highlighted in bold in the molecular formulae.

The resulting LLS lifetimes are reported in Table 1. Note that at both fields (11.7 T and 7 T), the ratios  $T_{LLS}/T_1$  all lie in the vicinity of 2. We see the pronounced effect of CSA on <sup>19</sup>F relaxation as both the  $T_I$  and  $T_{LLS}$  are longer at a field of 7 T (282.4 MHz for <sup>19</sup>F) for molecules PFBS and PFPeA. The ratios  $T_{LLS}/T_1$  remain rather similar at both fields, indicating both  $T_I$  and  $T_{LLS}$  are impacted by relaxation induced by CSA in a similar way. However, a clear trend in the change of  $T_I$  and  $T_{LLS}$  lifetimes measured at the two fields is not seen for PFBA. Fluorinated CF<sub>3</sub> methyl groups do not contribute to the LLS, as proven by selective decoupling at their resonance frequencies which affects neither the efficiency of LLS excitation of CF<sub>2</sub> groups nor their lifetimes.

155

Table 1: Optimized radio-frequency field amplitude  $v_{SLIC}$  and optimized duration  $\tau_{SLIC}$  for Spin-Lock Induced Crossings (SLIC) to generate long-lived states involving 4 or 6 <sup>19</sup>F spins of the fluorinated aliphatic chains, using 2 or 3 RF fields applied simultaneously to the 2 or 3 multiplets of the three fluorinated molecules (Fig. 2) at 11.7 T (470.46 MHz for <sup>19</sup>F) and 7 T (282.4 MHz for <sup>19</sup>F). The spin-lattice relaxation times  $T_1$  were determined by the inversion-recovery method. The relaxation times  $T_{LLS}$  were determined as described in Fig. 1b combined with a 4-step phase cycle (Sonnefeld et al., 2022b). The errors correspond to one standard deviation. The samples of PFBA, PFBS and PFPeA had concentrations of 500 mM, all in DMSO-d6.

| Compound                                | v <sub>SLIC</sub> [Hz] | τ <sub>SLIC</sub> [ms] | Peak | T <sub>1</sub> [s] at 11.7 T<br>(500 MHz) | T <sub>LLS</sub> [s] at 11.7 T<br>(500 MHz) | Ratio T <sub>LLS</sub> /T <sub>1</sub><br>at 11.7 T | T <sub>1</sub> [s] at 7 T<br>(300 MHz) | T <sub>LLS</sub> [s] at 7 T<br>(300 MHz) | Ratio T <sub>LLS</sub> /T <sub>1</sub><br>at 7 T |
|-----------------------------------------|------------------------|------------------------|------|-------------------------------------------|---------------------------------------------|-----------------------------------------------------|----------------------------------------|------------------------------------------|--------------------------------------------------|
| Perfluorobutanoic<br>acid (PFBA)        | 574                    | 60                     | 2    | 0.96 ± 0.01                               | 3.75 ± 0.14                                 | 3.4                                                 | 1.67 ± 0.02                            | 3.17 ± 0.25                              | 1.9                                              |
|                                         |                        |                        | 3    | 1.62 ± 0.02                               | 3.66 ± 0.16                                 | 2.1                                                 | 2.41 ± 0.01                            | 3.74 ± 0.23                              | 1.6                                              |
| Perfluorobutane<br>sulfonic acid (PFBS) | 576                    | 35                     | 2    | 1.26 ± 0.0                                | 2.78 ± 0.07                                 | 2.2                                                 | 2.32 ± 0.02                            | 5.14 ± 0.09                              | 2.2                                              |
|                                         |                        |                        | 3    | 1.21 ± 0.0                                | 2.92 ± 0.07                                 | 2.4                                                 | 2.22 ± 0.02                            | 4.56 ± 0.11                              | 2.1                                              |
|                                         |                        |                        | 4    | 1.27 ± 0.0                                | 2.91 ± 0.07                                 | 2.3                                                 | 2.31 ± 0.02                            | 4.59 ± 0.14                              | 2.0                                              |
| Perfluoropentanoic<br>acid (PFPeA)      | 585                    | 80                     | 2    | 0.69 ± 0.0                                | 2.24 ± 0.06                                 | 3.2                                                 | 1.2 ± 0.01                             | 3.28 ± 0.25                              | 2.7                                              |
|                                         |                        |                        | 3    | 1.04 ± 0.0                                | 2.23 ± 0.05                                 | 2.1                                                 | 1.59 ± 0.02                            | 3.11 ± 0.32                              | 2.0                                              |
|                                         |                        |                        | 4    | 1.09 ± 0.0                                | 2.41 ± 0.07                                 | 2.2                                                 | 1.7 ± 0.02                             | 2.8 ± 0.24                               | 1.6                                              |

## 2.3 Protein-Ligand Titration of PFBS with BSA

165

170

Figure 5.  $T_{LLS}$  and  $T_I$  contrast curves for eight samples with [PFBS] = 50 mM, and  $0 < [BSA] < 100 \,\mu\text{M}$ . The curves showing  $T_{LLS}$  contrast are shown for each of the CF2 groups while the  $T_I$  contrast is only shown for the central CF2 group. The contrast is normalized and expressed as a percentage between 0 and 100 %. The samples are composed of PFBS, BSA and phosphate buffer to obtain solutions of pH  $\approx 7.2$ .

180

185

190

195

It has been previously reported that PFAS have a binding affinity for various proteins (Zhao et al., 2023). To evaluate the ability of <sup>19</sup>F LLS to provide contrast between a free and bound form, when interacting with a protein, we explored the binding of PFBS to the protein Bovine Serum Albumin (BSA), by measuring  $T_{LLS}$  of samples with [PFBS] = 50 mM and variating concentrations of  $0 < [BSA] < 100 \mu M$ . The contrast can be defined as usual by  $C_1 = (R^{obs}_1 - R^{free}_1) / R^{obs}_1$  for experiments measuring longitudinal relaxation, with  $R_1 = 1 / T_I$ , or  $C_{LLS} = (R^{obs}_{LLS} - R^{free}_{LLS}) / R^{obs}_{LLS}$ , with  $R_{LLS} = 1 / T_{LLS}$ , for experiments measuring long-lived states lifetimes.

As shown in Fig. 5, the relaxation times,  $T_{LLS}$ , are clearly affected by binding of the ligand to the target protein. In addition, <sup>19</sup>F LLSs created in PFBS offer good contrast even if the protein is  $10^{-4}$  times more dilute than the ligand. Note that the contrast obtained with  $T_L$  (<sup>19</sup>F) is much worse than the contrast obtained with  $T_{LLS}$  (<sup>19</sup>F). We believe that the contrast can be improved at lower magnetic fields (where the chemical shift anisotropy is less efficient) but have not yet been able to verify this expectation.

### 3 Conclusion

It has been shown that for fluorinated aliphatic chains, the amplitudes of forbidden Outer Singlet-Triplet transitions (OSTs) can be boosted in one-dimensional  $^{19}F$  NMR spectra, which is crucial for the optimization of SLIC parameters to create long-lived states involving 4 or 6  $^{19}F$  spins in isotropic solution. We demonstrate that long-lived states of  $^{19}F$  spins can be readily excited in three different achiral molecules containing fluorinated aliphatic chains. At a field of 11.7 T (470.46 MHz for  $^{19}F$ , 500 MHz for  $^{1}H$ ), the lifetimes  $T_{LLS}(^{19}F)$  of the long-lived states exceed the longitudinal relaxation times  $T_1(^{19}F)$  by a factor between 2.1 and 3.4. We also measured  $T_1(^{19}F)$  and  $T_{LLS}(^{19}F)$  lifetimes at a field of 7 T (282.4 MHz for  $^{19}F$ , 300 MHz for  $^{1}H$ ) which were overall longer than those measured at 11.7 T, but showed similar ratios,  $T_{LLS}/T_1$ . Fluorinated aliphatic chains can be attached to existing drugs that have metabolic or pharmacokinetic properties. This makes the excitation of LLS interesting for drug screening, where it has been demonstrated that LLS show good contrast when binding to target proteins.

#### **Author Contributions**

K.S. designed the research. C.W. and S.V.D. performed the experiments and analysed the data. All authors contributed to writing the paper.

#### **Conflict of Interest**

G.B is a member of the editorial board of Magnetic Resonance of the Groupement Ampere. The peer-review process was guided by an independent editor, and the authors have no other competing interests to declare.

## **Financial Support**

This work was supported by the European Research Council (ERC), Synergy grant "Highly Informative Drug Screening by Overcoming NMR Restrictions" (HISCORE, grant agreement number 951459). K.S. acknowledges support by l'Agence Nationale de la Recherche (ANR) on the project THROUGH-NMR (ANR-24-CE93-0011-01).

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
