# Peer review of "Long-Lived States Involving a Manifold of Fluorine-19 Spins in Fluorinated Aliphatic Chains"

_Magnetic Resonance, 2025_

## Author Comment (AC1)

Paris, September 24th 2025

Re: "Long-Lived States Involving a Manifold of Fluorine-19 Spins in Fluorinated Aliphatic Chains", by Coline Wiame, Sebastiaan Van Dyck, Kirill Sheberstov, Aiky Razanahoera and Geoffrey Bodenhausen

Dear editor,

Please find our responses below.

**Reviewer 1:**

The manuscript by Wiame, et al. presents a first analysis of long-lived $^{19}$F spin order in fluoroalkyl chains, building on the group's previous works investigating long-lived $^{1}$H spin order in aliphatic chains such as Sonnefeld et al. 2022, https://doi.org/10.1103/PhysRevLett.129.183203 ibid 2022, https://doi.org/10.1126/sciadv.ade2113, and Sheberstov et al, 2024 https://doi.org/10.1063/5.0196808.  Unfortunately, I feel that compared to those impressive works, the presentation of the current manuscript is uneven and, in many places vague, clumsy and confusing, and requires substantial improvement.

It is very strange to me that the abstract claims the interesting results

"The lifetimes TLLS(19F) can be strongly affected by binding to macromolecules, a feature that can be exploited for screening of fluorinated drugs".

And

"TLLS(19F) and T1(19F) should be longer at low fields where relaxation due to the chemical shift anisotropy (CSA) of 19F is less effective, as will be shown elsewhere"

but presents zero results about either of these in the current manuscripts.  I would strongly discourage this sort of "splitting results" across two papers.  It seems to me like the authors want to publish two or even more papers where they should really only be publishing one.

We thank the reviewer and we have added experimental results (Fig. 5) showing binding to protein. We removed the premature statement about CSA.

I would like to emphasize this using the most positive aspect of the paper - that the authors measured collective-$^{19}$F singlet lifetimes that are longer than the $^{19}$F $T_1$ by around a factor of 3, which might indeed be useful in some applications. But the conclusions section begs the question what the relaxation times are at other magnetic fields. Even if you went to a 300 or 400 MHz spectrometer, the lifetimes would be significantly different if there were a major contribution from chemical shift anisotropy (where the rate contribution is proportional to field squared)

If the authors already have these results in hand, as stated in the abstract, it would be poor conduct not to discuss these – at the very least, in some preliminary form – in the present paper directly.

So far, we don't have any results about the field-dependence of relaxation.

Below is some more detailed criticism that I hope allows the authors to improve their manuscript.

1. The text lacks detail in many places. In many places, as detailed below, the authors' claims are vague and to make any sense to readers these deserve explanation or extra qualification/quantitation.

Line 24, "The absence of background signals offers decisive advantages (Buchholz and Pomerantz)". Please help the reader by spelling out what these "decisive" advantages are.

Thank you for the comment. We changed the sentence to be: "Fluorinated drugs also have the advantage of being easy to track by NMR and MRI, since the sensitivity of $^{19}$F NMR is comparable to that of $^1$H NMR, while signal overlap is significantly reduced due to the greater chemical shift dispersion, and spectral quality is improved because of the absence of background signals, e.g., water signal (Buchholz and Pomerantz, 2021)."

Line 26, "Good contrast can be achieved in 19F NMR". What is "good contrast"? Good is not an objective word.

The revised sentence now reads as following: "For ligands that bind to a target protein, contrast in $^{19}$F NMR arises from the combined effects of binding on the chemical shifts, on the reduction of symmetry when achiral ligands bind to chiral targets, and on the correlation times of rotational diffusion."

Line 46, "These remarks apply to many molecules". Please say what these are, give a few examples.

Here is the revised sentence: "These remarks apply to molecules containing either three fluorinated or three protonated methylene groups, including those analyzed in this study and in our previous work on protons (Sonnefeld et al., 2022a)."

Line 88, "different symmetries of the spin permutation group". Can you elaborate a bit further on this? I think it is only a specific type of symmetry of the transition dipole operator that leads to long lived spin order, rather than "different" symmetries, so you can comment more explicitly about that.

Thank you for the question. We actually meant a specific example of LLS in CD3 groups. The sentence now reads: "In systems with more than two spins, one can also excite long-lived imbalances between states that belong to different symmetries of the spin permutation group, e.g. an imbalance between populations associated with irreducible representations A and E in $CD_3$ groups (Kress et al., 2019)."

Line 97, "their chemical shifts cover a sufficient spread". There is no indication of what "sufficient" refers to. Needs clarification, preferably use alternative wording.

Thank you for pointing this out. "Sufficient spread" means that the LLS excitation of one $CF_2$ group must be sufficiently selective, e.g., that one can neglect the effects of a SLIC pulse acting on a neighboring $CF_2$ group. The typical RF amplitude of a SLIC pulse is on the order of 500 Hz for $^{19}F$, so that the chemical shifts difference should be at least 1 kHz.

The paragraph now reads as follows: "We have explored three achiral fluorinated molecules shown in Fig. 2, selected because the chemical shifts between neighboring $CF_2$ groups exceed 1 kHz. This means that LLS excitation of a chosen $CF_2$ group is sufficiently selective, even with RF amplitudes on the order of 250 to 290 Hz for DQ LAC, or on the order of 500 to 580 Hz for SQ LAC respectively, so that one can neglect the effects of a SLIC pulse on neighboring $CF_2$ groups."

Line 120, "These OSTs are much harder to detect... because the scalar couplings are more favorable" is a sentence that I can appreciate the meaning of. However, it is again not objective, and will be nearly impossible to understand for most readers.

Thank you for pointing this out. After some rethinking we removed this sentence, as in fact we have also encountered examples of molecules with protons in $CH_2$ chains where the intensity of OST transitions is negligibly small.

There are also a few very long and cumbersome sentences that need to be rewritten. Ideally as several sentences, to get the point across clearly. For example

Line 29, "The latter two effects can be distinguished by comparison of, on the one hand, conventional Carr-Purcell-Meiboom-Gill (CPMG) echo trains with high repetition rates required to suppress echo modulations due to homonuclear scalar couplings nJ(19F, 19F),

which also inhibit echo decays due to intermediate exchange, and, on the other hand, slow CPMG echo trains using the so-called "perfect echoes" to eliminate the effects of nJ(19F, 19F) couplings while retaining the effects of intermediate exchange (Takegoshi et al., 1989; Aguilar et al., 2012; Lorz et al., 2025)."

Yes, that is one sentence.

This lengthy sentence can be broken up as follows:

"The latter effect leads to line-broadening by homogeneous transverse $T_2$ relaxation. In addition, chemical exchange in the intermediate regime can also contribute to line-broadening (Buchholz and Pomerantz, 2021). The latter two effects can be distinguished by comparing Carr-Purcell-Meiboom-Gill (CPMG) echo trains with high repetition rates and slow CPMG echo trains using so-called "perfect echoes". The first type of experiment will suppress echo modulations due to homonuclear scalar couplings $^nJ(^{19}F, ^{19}F)$ and inhibit echo decays due to intermediate exchange (Takegoshi et al., 1989; Aguilar et al., 2012), while the second type of experiment eliminates the effects of $^nJ(^{19}F, ^{19}F)$ couplings while retaining the effects of intermediate exchange (Lorz et al., 2025)."

Plus

Line 59, "The degeneracy of vicinal scalar couplings is lifted provided the potential wells of the different rotamers that result from rotations about the C-C bonds are not equally populated."

This is shorter, but still more convoluted than it needs to be.

We rephrased it: "The degeneracy of vicinal scalar couplings is lifted when the rotamers produced by rotations about the C–C bonds are not equally populated. This occurs if the potential wells corresponding to the different rotamers have unequal energy."

1. I have many comments on scientific rather than grammatical issues:

Line 16 (Introduction), the published literature also contains examples of long-lived spin order between $^{15}N_2$, $^{31}P$ and 103Rh spin pairs, which I think deserve a mention along with $^1H$ and $^{13}C$. A mention of dihydrogen (or specifically ortho or para-enriched $H_2$) is also notably absent.

We have mentioned *ortho* or *para*-$H_2$ in the introduction, and mentioned $^{15}N_2$, $^{31}P$ and $^{103}Rh$ spin pairs in the introduction:

"The discovery of long-lived states (LLS) was inspired by spin isomers of dihydrogen ($H_2$), in which a nonequilibrium ratio of para- vs ortho-dihydrogen can persist for many days before relaxing. Many applications of LLS have been illustrated for pairs of $^{13}C$ or $^{15}N$ nuclei (Pileio et al., 2012a; Feng et al., 2013; Stevanato et al., 2015; Elliott et al., 2019; Sheberstov et al.,

2019c), for pairs of $^1$H nuclei (Franzoni et al., 2012; Kiryutin et al., 2019a), and less frequently, for pairs of $^{19}$F nuclei (Buratto et al., 2016), for $^{31}$P (Korenchan et al., 2021), and for $^{103}$Rh spins (Harbor-Collins et al., 2024). The slow decays of LLS have been exploited for the determination of small diffusion coefficients (Cavadini et al., 2005), for measurements of slow chemical exchange (Sarkar et al., 2007), for the storage of hyperpolarization (Vasos et al., 2009; Pileio et al., 2012; Kiryutin et al., 2019), for investigations of weak ligand-protein binding (Salvi et al., 2012), and for studies of tortuosity in porous media (Dumez et al., 2014; Pileio et al., 2015; Pileio and Ostrowska, 2017; Tourell et al., 2018; Melchiorre et al., 2023). Another application of LLS is spectral editing, in particular for selective filtering of selected signals in proteins (Mamone et al., 2020; Sicoli et al., 2024)."

Line 22, Fluorinated drugs can in some cases be synthesized with a 18F radiolabel and their biodistribution studied using PET scanners. An important application that the authors could mention.

We have added the following sentence: "Positron emission tomography (PET) with $^{18}$F-labeled radioligands is widely applied in diagnosis and in drug development (Nerella et al., 2022)."

Line 46, "mono-exponential decays" should be "an effective mono-exponential decay".

We agree and changed the text to read:

"...tend to relax with an effective mono-exponential decay."

Line 64, "PFAS molecules..., which have been widely studied for the detrimental effect on the environment". It seems backwards to perform a study on a molecule that has negative environmental effects. I don't think this (as the sentence currently reads) can be claimed as a motivation or justification for the molecules studied.

We restated the text as following: "Specifically, we have looked at per- and polyfluoroalkyl (PFAS) molecules, which have been widely studied for their detrimental effects on the environment (Fenton et al., 2021). Studying these molecules and their ability to bind to protein targets could shed more light on their impact on living organisms."

Line 83, "T00 filter" is missing a reference. A general one is the chapter "Filters for long-lived spin order" https://doi.org/10.1039/9781788019972-00188

Thank you, we added the reference.

Line 91, "J(19F-19F) couplings obey different rules compared to J(1H-1H) couplings". I don't agree with this sentence. Physically speaking, a J coupling is a J coupling and there are no "rules" that depend on the nuclear spin species involved. I think the authors instead mean

"J(19F-19F) and J(1H-1H) couplings differ in the dependence of their magnitude on the number of chemical bonds between atoms".

We agree and changed the sentence accordingly.

Line 119, "(henceforth called Outer singlet-triplet transitions or OSTs)" introduces this terminology late in the paper, where it is already used two pages earlier in the caption of Figure 1. For cleanliness please introduce this terminology earlier, before referring to Figure 1.

We added the following sentence before introducing Fig.1: "Excitation with a SLIC pulse can also enhance the intensity of Outer Singlet–Triplet (OST) coherences (Sheberstov et al., 2019a), as discussed in more detail below."

Line 125, "Once the amplitude vSLIC has been optimized, the duration τSLIC can readily be optimized by searching empirically for the highest signal amplitude." Similar optimization procedures are used for pulse sequences such as M2S or PulsePol-type sequences that excite OST states. The authors might want to refer to those too.

Sure, we added the following sentence: "An enhancement of OST transitions can also be achieved using other techniques, such as J-synchronized CPMG (Sheberstov et al., 2019a) or symmetry-based sequences (Sabba et al., 2022)."

Line 131, "forbidden combination lines" should read "weakly allowed combination lines".

We changed the text accordingly.

Line 143 (caption of Figure 4) is sloppy, reading "actually admixtures of two-, four-, and minor amounts of six-spin order LLS terms". Alternative, more formal wording is needed here.

Although we do not think our wording was 'sloppy', we have revised the caption: "Decays of LLS (actually admixtures of two-, four-, and minor amounts of six-spin order LLS terms) for three different fluorinated molecules."

However, I am generally not convinced about including the Figure in the paper because all of the important information can be found instead in Table 1.

The figure allows one to appreciate the quality of the data.

If the authors do want to keep the figure, then perhaps plotting on a semi-log graph (with error bars) would be a better way to show that the decay curves are close to monoexponential.

The experimental points are overlayed with monoexponential decay curves, so it should be clear from the figure.

**Reviewer 2:**

In this work, authors excite and measure the lifetimes of long-lived states (LLS) of 19F spins in three fluorinated aliphatic molecules and show that the LLS time constants are 2.1 to 3.6 times longer than the T1 time constants. They claim that LLS time constants are sensitive to binding partners, in which case this study will be interesting. While this claim is supported by previous literature (Buratto et al., 2016, J. Med. Chem.), this manuscript does not contain any studies with binding molecules.

We have inserted new material showing experiments with protein binding.

This work may be interesting for those who want to take it further for actual applications. The authors should address the following questions and make appropriate revisions before proceeding further.

Questions

1. What challenges are faced for excitation of LLS in the case of 19F instead of 1H?

The main challenge results from the range of amplitudes of the relevant couplings, which are larger by an order of magnitude for $^{19}$F. It is therefore necessary to perform optimization of SLIC pulse more carefully. The paragraph describing this is the following:

"To extend the excitation of LLS from $^{1}$H to $^{19}$F, a challenge arises from the fact that $J(^{19}$F-$^{19}$F) and $J(^{1}$H-$^{1}$H) differ in the way their magnitude depend on the number of chemical bonds between atoms. Typically, geminal $^{2}J(^{19}$F-$^{19}$F) couplings in $CF_2$ groups are on the order of 250 to 290 Hz (Krivdin, 2020), much larger than geminal $^{2}J(^{1}$H-$^{1}$H) couplings in $CH_2$ groups which are about ~ -15 Hz. In $^{1}$H NMR, typical values of the difference between vicinal couplings lie in the range $0 < \Delta J(^{1}$H$) < 7$ Hz, while for $^{19}$F NMR, these differences may typically lie in the range $0 < \Delta J(^{19}$F$) < 40$ Hz."

2. In the final state, what are the relative proportions of different spin order terms (2, 4, and 6)?

The 6-spin terms are typically very weak, only a few percent of the 2- and 4-spin terms

Unfortunately, we cannot exactly calculate the proportions of the different spin terms because we do not have enough information on the coupling constants between the $^{19}$F nuclei in the various molecules studied.

3. From the optimal SLIC parameters, how to estimate the F-F J-couplings?

The answer is given in the caption to Figure 1:

"The RF amplitudes must be twice the geminal coupling $n_{SLIC}^{SQ} = 2\,^{2}J(^{19}$F, $^{19}$F) for SQ LAC, or equal to the geminal coupling $n_{SLIC}^{DQ} = {}^{2}J\,(^{19}$F, $^{19}$F) for DQ LAC. The maximum efficiency is

achieved either for a short pulse duration $\tau^{SQ}_{SLIC} = 1/(|\sqrt{2}\Delta J|)$ for SQ LAC or a longer duration $\tau^{DQ}_{SLIC} = \sqrt{2}\tau^{SQ}_{SLIC} = 1/(|\Delta J|)$ for DQ LAC, where $DJ = (DJ_{AM} + DJ_{MX})/2$ and $DJ_{AM} = J_{AM} - J_{AM'} = J_{A'M'} - J_{A'M}$ and $DJ_{AX} = J_{AX} - J_{AX'} = J_{A'X'} - J_{A'X}$."

4. Why was the first F-F pair not considered in all three cases?

We considered only LLS in fluorinated *methylene* groups. The numbering of the atoms starts with a $CF_3$ methyl group for which there are no accessible LLS because the three $^{19}F$ nuclei are magnetically equivalent.

5. Will this method extend to low fields and strong coupling cases?

So far, we have not been successful in our attempts to excite LLS of the molecules studied in this paper in a benchtop NMR at 1.4 T.

We hope that our revisions will be deemed sufficient by the reviewers.

Sincerely,

Kirill Sheberstov

---

## Author Response (AR2)

Paris, October 7th 2025

Re: "Long-Lived States Involving a Manifold of Fluorine-19 Spins in Fluorinated Aliphatic Chains", by Coline Wiame, Sebastiaan Van Dyck, Kirill Sheberstov, Aiky Razanahoera and Geoffrey Bodenhausen

Dear editor,

Please find our responses below.

**Reviewer 1:**

I consider it suitable for publication after some corrections / modifications:

- Even if a deeper study on CSA relaxation effects is the objective of the authors for another paper, I consider it disappointing for the CSA discussion (and request for data) to have been dropped completely from this paper. As someone interested in relaxation mechanisms, beyond learning from the paper that TLLS<T1 in this class of molecules, I really would like to have some more of a hint "why". Assuming the authors do have an extensive NMR department with access to a say a 300/400 MHz NMR, in addition to the 500 MHz spectrometer used in this work, a valuable addition to this story seems low-hanging fruit: a quick measurement of TLLS/T1 for one molecule at two fields would be enough.

We thank Michael for your persistence; we performed measurements at 300 MHz spectrometer. These results are now incorporated into the revised version of manuscript. Indeed, both T1 and TLLS become slightly longer at the lower field, revealing a slight contribution from CSA.

- In Figure 5, the dataset for the binding-affinity study, I had to search for the meaning of "contrast" on the vertical axis of the plot. Eventually I found an equation in the main text (paragraph starting line 225) expressing "C\_LLS" in terms of R\_LLS=1/T\_LLS values with/without protein in solution. Please make this more explicit in the figure by doing the

following: (1) Label the Y axis as "C\_1, C\_LLS", (2) State directly that C is a normalized contrast that is bounded between 0 and 1, (or 0 and 100%), assuming R\_LLS in the presence of the protein is faster than the baseline (are there any situations where negative contrast is expected?). (3) State the analogous equation for C\_1, in terms of T\_1.

Thank you, we followed your suggestions improving the figure, mentioning the normalization in the caption, and providing the equation for T1.

Sincerely,

Kirill Sheberstov